# Use of Hazelnut Perisperm as an Antioxidant for Production of Sustainable Biodegradable Active Films

**DOI:** 10.3390/polym14194156

**Published:** 2022-10-04

**Authors:** Paola Scarfato, Maria Luisa Graziano, Arianna Pietrosanto, Luciano Di Maio, Loredana Incarnato

**Affiliations:** Department of Industrial Engineering, University of Salerno, Via Giovanni Paolo II, 132, 84084 Fisciano, SA, Italy

**Keywords:** active packaging, hazelnut perisperm, biodegradable film, antioxidant activity, PLA/PBAT blends, food packaging

## Abstract

Utilization of food-waste-derived bioactive compounds with biodegradable polymers is an attractive strategy leading innovation in the food packaging sector and contributing to reduce the environmental concerns of plastic packaging disposal. In this field, this work is aimed to use hazelnut perisperm as an antioxidant agent in the production of biodegradable polymeric films for active packaging applications. For this purpose, hazelnut perisperm of a selected particle size (<250 μm) at different percentages (0%, 5% and 10% by weight) was added to a bioderived and compostable polymer suitable for food contact, known as Ecovio^®^. The blends were produced by a twin-screw extrusion process, while active films were prepared with a pilot lab-scale film blowing plant. The films were characterized in terms of physical–mechanical properties (thermal, tensile, oxygen barrier, optical, sealing ability) and antioxidant activity (DPPH), to investigate their potential use as active packaging. The results showed that the presence of the hazelnut perisperm confers significant antioxidant activity to the films, which is useful in counteracting lipid oxidation and preserve the quality of lipophilic foods, e.g., nut-dried fruits. An extension of the sealability temperature range of the films without compromising their strength was also highlighted. Moreover, the hazelnut perisperm causes a gradual decrease in the stiffness and mechanical strength of the films and an increase in the ductility of the system.

## 1. Introduction

The growing attention to environmental sustainability has led to overcoming the linear economic model in favor of a circular economy model capable of being self-sustaining and self-regenerating. The valorization of waste streams and by-products coming from agri-food chains is fully in line with the circular economy, as what represents the waste of one process becomes the raw material for another one [1,2,3,4,5,6].

Among the various agri-food chains, hazelnut tree (*Corylus avellana* L.) cultivation represents a relevant crop in the Italian national agricultural scenario, with an annual production of hazelnut seeds of about 110,000 tons, which makes Italy the second largest producer of hazelnuts in the world, after Turkey [7]. One of the wastes of this industry is represented by the hazelnut perisperm, a dark brown skin with a bitter-astringent taste, which constitutes about 2.5% of the total weight of the fruit and is generally discarded at the time of roasting. However, this skin is an extremely rich source of fibers and polyphenols, i.e., compounds with antioxidant action [8,9,10,11].

In the search to find valuable strategies to recover these high-value ingredients, a lot of research was devoted to detection and comparison of bioactive compounds in different extracts of hazelnut skin [12,13,14,15]. All these studies showed that the extracts obtained from the perisperm of hazelnuts have excellent reducing properties and excellent radical-scavenging activities, generally superior to those found in the most common synthetic and natural antioxidants used commercially. Therefore, they have great potential in several industrial fields as functional food additives, nutraceuticals, therapeutics, cosmetics, etc. [16,17,18].

Recently, driven by the need for a more rational use of resources and by the growing demand for more sustainable packaging solutions, some authors have also explored the potential applications of hazelnut industry wastes and hazelnut skin extracts as natural additives for eco-friendly and active packaging [19,20,21]. Battegazzore et al. (2014), through a multi-step extraction process, obtained three fractions of hazelnut skins extracts, which they used not only for their antioxidant activity, but also as plasticizers and reinforcement fillers for poly (lactic acid) (PLA) and polypropylene (PP) [20]. The extracts/polymer blends were melt compounded using a twin screw micro extruder and then characterized. The antioxidant fraction turned out to be effective as a UV absorber and thermal stabilizers for PP, while the other fractions were found capable of plasticizing PLA and increasing the storage modulus of PLA and PP. In another study, Esposito et al. (2020) reported on fabrication via a solvent casting technique of bioactive pullulan-based films loaded with roasted hazelnut skins extracts [21]. The films, tested for antibacterial activity towards Staphylococcus aureus, showed effective antibacterial activity, reaching a maximum of 4–log CFU mL^−1^ reduction for the better performing formulation at 10 wt% of extract content. However, to the best of our best knowledge, there are no studies regarding the incorporation of the whole hazelnut perisperm in polymeric matrices to produce biodegradable active films for the extension of food shelf life by conventional polymer processing technologies.

Thus, this work investigates the effectiveness of hazelnut perisperm as a functional agent for the development of innovative, fully bioderived active packaging films, which are: (i) suitable for direct food contact, and (ii) reusable in the agri-food supply chains, from a circular economy perspective. To this aim, hazelnut perisperm was firstly powdered and melt compounded at two different loading levels (5 and 10 wt%) in a commercial bioderived and biodegradable polymer matrix (Ecovio^®^, by BASF) based on poly (butylene adipate-co-terephthalate) (PBAT) and PLA. This matrix was selected based on literature findings indicating that PLA is one of the most suitable packaging materials to preserve quality of nut-dried fruits [22]. The blends were used to produce films in different processing conditions on a pilot-scale film-blowing extrusion plant. Then, the films were analyzed for their morphology, thermal, mechanical, optical and gas transport behavior, water wettability, sealing ability and antioxidant activity, to evaluate their potential as active food packaging.

## 2. Materials and Methods

### 2.1. Materials

Hazelnut perisperm (HP) from Tonda di Giffoni variety was obtained by a local producer from Giffoni Valle Piana (Salerno, Italy) after roasting hazelnuts at 150 °C for 25 min. The polymer matrix is the Ecovio F2332 (BASF SE, Ludwigshafen, Germany), a commercial biodegradable blend based on poly (butylene adipate-co-terephthalate) and polylactid acid, whose main properties are reported in Table 1 [23].

### 2.2. Active Film Preparation

In order to obtain the mixtures with a suitable dispersion of the perisperm in the polymeric matrix, it was necessary to optimize the average size of the perisperm particles through preliminary tests, reducing them with the use of a grinder working with liquid nitrogen. The powders were passed through a sieve of size <0.25 mm (Figure 1), added at 5% and 10% by weight to Ecovio polymer pellets and then mechanically mixed with a matrix and dried under vacuum for about 24 h at 80 °C prior to any use to avoid thermal hydrolytic degradation during the extrusion.

The dry blends were melt-mixed by a Collin ZK25 modular twin-screw extruder [8] equipped with co-rotating interpenetrating screws (L/D = 42, D_screw_ = 25 mm) and a gravimetric dosage system. The extrusion was performed at a feeding rate of 0.9 kg/h, screw speed equal to 180 rpm and with the following thermal profile (from the hopper to die): 140–175–180–180–180–180–180–175 °C. The molten stream was collected in a water bath, pelletised by a pelletizer and used for the film production, after being dried under vacuum for 24 h at 80 °C.

Films were produced by an extrusion film-blowing process by making use of a laboratory scale plant (GIMAC) single-screw extruder (L/D = 24, D_screw_ = 12 mm). The film production was performed at screw speed equal to 35 rpm and a temperature profile (from hopper to die): 170–160–150 °C. The films were blown up to a bubble diameter of 50 mm (corresponding to a band width of ca. 80 mm) and collected at three different draw-up speeds: 0.9, 1.6 and 3 m/min, thus allowing us to obtain samples with different thicknesses. In all cases, the process was quite stable as also shown in Figure 2. Using the same procedure, pure Ecovio films were also produced for comparison. Sample nomenclature and characteristics are specified in Table 2.

### 2.3. Methods

Hazelnut perisperm was analyzed both not dried (HP) and after drying (HP dried) at 80 °C for 16 h.

Fourier transform infrared spectroscopy measurements in attenuated total reflectance mode (ATR−FTIR) were performed with a NEXUS 600 spectrophotometer equipped with the Smart Performer accessory (Thermo Fisher Scientific, Waltham, MA, USA). The spectra were collected in the 4000–600 cm^−1^ frequency range with a resolution of 2 cm^−1^ and averaged on 64 scans.

Differential Scanning Calorimetry (DSC) analysis was carried out with a DSC mod. 822 (Mettler Toledo, Columbus, OH, USA) under a nitrogen flow (100 mL/min) to avoid oxidative phenomena. Three scans were performed: samples were firstly heated from −70 to 300°C with a scan rate of 10 °C/min, and held at 300 °C for 5 min; then, they were cooled to −70 °C at 10 °C/min and finally heated again to 300 °C at 10 °C/min.

Thermogravimetric analysis was performed by a TGA/SDTA851 (Mettler Toledo, Columbus, OH, USA). The samples (about 7 mg) were heated from 25 °C to 800 °C at a rate of 10 °C/min under an inert atmosphere (N_2_ flow = 20 mL/min).

The 2,2-diphenyl-1-picrylhydrazyl (DPPH) radical scavenging assay was applied to both hazelnut perisperm (as-is and dried) and active films to evaluate their radical scavenging activity. The scavenger activity was evaluated in terms of extent of the DPPH radical reduction reaction given by the active agent, through spectrophotometric measurements at 515 nm where the DPPH has its maximum of absorbance. To perform the DPPH test on the as-is and dried perisperm, antioxidant extracts were obtained from 0.1 g of perisperm by soaking them in 2 mL of 95% *v*/*v* ethanol and storing them in the dark for 1 week. Instead, to examine the radical scavenging activity of the active films, antioxidant extracts were obtained by immersing 1 dm^2^ of film in a plate containing 100 mL of 95% *v*/*v* ethanol and shaking it on a plate shaker for 20 days to allow the complete release of the active agents from the films. A stock solution was made with 24 mg of DPPH radical solubilized in 100 mL of ethanol. The working solution was then prepared with 10 mL of stock solution and 90 mL of ethanol. Then, 1950 μL of the working solution was mixed with 50 μL of ethanolic antioxidant extracts (pure ethanol was used as blank) and left in the dark for 20 min at room temperature. Absorbance was measured at 515 nm using a UV-Vis spectrophotometer (Lambda 800, Perkin Elmer, Waltham, MA, USA). The percentage radical scavenging activity (RSA%) was calculated as a percentage of DPPH radical discoloration according to the following equation:RSA% = (A_control_ − A_sample_)/(A_control_) × 100(1)
where A_control_ and A_sample_ are the control absorbance and the sample absorbance, respectively. The reported results were averaged on three replicates of each sample.

Optical microscopy (OM) observations were performed on film surfaces by means of a Zeiss Axioskop microscope (Carl Zeiss Vision GmbH, Aalen, Germany) operating in reflection mode.

Mechanical tensile properties were determined using a SANS dynamometer (mod. CMT 6000 by MTS, Shenzhen, China), equipped with a 100N load cell. The tests were performed in the machine direction (MD) on rectangular specimens (12.7 mm × 40 mm), according to ASTM D822. The values of the reported mechanical parameters were the average of at least 10 replicates for each sample.

Oxygen permeability measurements were performed by means of a gas permeabilimeter (GDP-C, Brugger, Munich, Germany). The tests were carried out in triplicate on samples having an area = 78.4 cm^2^, at 23 °C and 0% R.H., under pressure difference of oxygen equal to 1 bar, with the oxygen flow rate of 80 mL/min according to the standard test method ISO 15105-1. The oxygen permeability values (PO_2_) were calculated as the oxygen transmission rates multiplied by the film thicknesses.

Ultraviolet–visible (UV–vis) spectroscopy measurements were carried out on film samples by means of a Lambda 800 UV-VIS spectrophotometer (Perkin Elmer, Waltham, MA, USA), according to the standard test method ASTM D1746. The transparency of the films measured as the percent transmittance at 560 nm was averaged on five replicate specimens.

Colorimetric analysis of the films was carried out with a CR-410 HEAD colorimeter (Konica Minolta Sensing, Inc., Nieuwegein, The Netherlands) based on the L*, a* and b* coordinates of the CIELAB space according to the Italian Recommendation NORMAL 43/93 [24]. Namely, the L* values range from 0 to + 100 and represent black and white, respectively, the negative and positive a* values represent green and red, respectively, while the negative and positive b* values represent blue and yellow. From the parameters above, the total color difference, ΔE, was calculated using the pure Ecovio D0.9 film as a reference, according to the formula [25]:(2)$$∆E=\sqrt[{2}]{{\left(∆L^{*}\right)}^{2}+{\left(∆a^{*}\right)}^{2}+{\left(∆b^{*}\right)}^{2}}$$
where the reported ΔE values were the mean value of ten measurements per area.

To evaluate the seal strength of the films, delamination tests were carried out by SANS dynamometer (mod. CMT 6000 by MTS, Shenzhen, China), equipped with a 100 N load cell, in accordance with the standards ASTM F88-00 and ASTM F2029-00. Welding of the films was carried out using a heat-sealing machine, Brugger model HSG-C, according to ASTM standard F 1921. Film strips 15 mm wide and 20 cm long were placed between two heated bars 15 cm × 1 cm in size and pressed at a force of 180 N for 1s at various temperatures (mod. HSG-C, Brugger, Munich, Germany). Specimens were sealed at different temperatures between 90 °C and 110 °C; these temperatures corresponded to the seal initiation temperature and the temperature above which the film suffers excessive distortion and shrinkage, respectively; the dwell time was set equal to 1 s, and the clamp force was set at 690 N. Next, according to the ASTM F88-00 standard; they were conditioned at 23 °C and 50 ± 5% R.H. for 48 h prior to testing. The bonding strength was evaluated in tensile mode, fixing a crosshead speed equal to 250 mm/min until seal failure. For each sample type, at least ten measurements were performed to assess the reproducibility of the results.

## 3. Results

### 3.1. Hazelnut Perisperm Characterization

The hazelnut perisperm was firstly characterized both in the “not dried” and “dried” state to investigate the possible occurrence of chemical structural changes due to the drying process required before extrusion with polymer materials sensitive to degradation by thermal hydrolysis, such as Ecovio^®^.

ATR/FT-IR analysis was carried out on the hazelnut perisperm to obtain structural information on their main constituents. Figure 3 compares the ATR−FTIR spectra taken on not dried (HP) and dried (HP dried) hazelnut perisperm samples.

The graph clearly shows that the two spectra are practically superimposable, thus indicating no perceivable changes in the chemical bonds of the hazelnut skin constituents after drying in the adopted experimental conditions. The main in characteristic peaks of the spectra are those related to lignin, which is the most abundant constituent of HP (37% by weight), according to the following assignment [26]: 3300 cm^−1^: stretching vibration of aliphatic and aromatic -OH; 2920 cm^−1^ and 2850 cm^−1^: stretching vibration of the C–H band in the CH_2_, CH_3_ and CH_3_O groups; 1740 cm^−1^: C=O stretching of acetyl; 1650 cm^−1^: C–O bonds; 1540 cm^−1^: aromatic skeletal vibration (C=C); 1450 cm^−1^: stretching of the phenol-ether bond; 1240 cm^−1^ and 1150-1025 cm^−1^: C–O and C–O–C stretching.

The DSC analysis was performed on the not dried and dried HP samples to obtain information on their morphology and thermal transitions. The thermograms of the two heating scans and the thermal parameters calculated in each scan of the thermal cycle are reported in Figure 4a,b and Table 3, respectively. In the thermogram relative to the first heating, two endothermic peaks are present. The first one, centered at about −8 °C, with a low melting enthalpy, is attributable to the melting of the oily fraction (which represents about 9% of the weight of the hazelnut perisperm) according to other authors [27], who reported that the melting temperature of hazelnut oily fraction is in the range from −10 °C to −5 °C. The second endothermic peak, larger and centered at about 90 °C is attributable to the evaporation of water, which represents about 8% of the mass content in hazelnut perisperm, and other volatile constituents [16].

In the cooling scan (curves not shown), there is only one exothermic peak at about −50 °C, related to the crystallization of the oily part, moisture and volatiles having now been removed. Then, these crystallized oily constituents melt in the second heating, giving a thermal transition at about −8 °C as in the first scan. These findings also show no perceivable changes in the thermal behavior of the hazelnut skin constituents from drying.

Thermogravimetric analysis was performed on not dried and dried HP hazelnut perisperm samples to evaluate their thermal stability and to fix processing conditions for blends preparation by melt compounding with the polymeric matrix. The obtained TGA curves and the corresponding thermal parameters (degradation onset temperature, T_onset_ and peak temperatures, T_I_–T_II_–T_III_–T_IV_) are reported in Figure 5 and Table 4, respectively.

The graph shows that both the hazelnut perisperm samples have a complex decomposition pattern consisting of several consecutive degradation steps. The first step around 100 °C is attributable to water elimination. As expected, the phenomenon starts at a higher temperature giving a lower weight loss for the HP dried (ca. 3%) than for the not dried one (ca. 9%) sample. This is because the residual moisture contained into the dried HP sample is more strictly bound to the other hazelnut perisperm constituents [16]. Then, the degradation of the hazelnut perisperm begins: the extrapolated onset temperature (To) is around 210 °C for both not dried and dried HP samples. This value represents the upper temperature limit that can be reached during the melt compounding with the polymer matrix to avoid perisperm degradation.

Finally, the DPPH assay was applied to examine the radical scavenging activity (RSA) of not dried and dried hazelnut perisperm samples. The results, reported in Table 5, shows that the drying process has not significantly affected the antioxidant activity of the perisperm. The small difference between the two samples can be due to removal of some volatile constituents as pointed out by DSC and TGA measurements.

### 3.2. Active Film Characterization

The films produced at different contents of HP (0%, 5% and 10%) have been characterized for their morphology, antioxidant activity and main relevant functional performances of interest for packaging applications (mechanical, oxygen barrier, optical and sealability).

Optical microscopy images of the Ecovio/HP 95/5 and Ecovio/HP 90/10 blend films produced at draw-up of 0.9 and 3 m/min are shown in Figure 6.

The pictures show that the filler is dispersed quite finely into the polymer matrix, but the distribution is not very homogeneous and became poorer both at increasing the HP loading and at decreasing the film thickness, to the point at which the Ecovio/HP 90/10 D3 film had big aggregates of about five hundred microns.

To investigate possible interactions between the polymer matrix and the HP filler, ATR−FTIR spectroscopic analysis was performed on all produced film. Figure 7 shows the spectra of pure Ecovio D0.9, Ecovio/HP 95/5 D0.9 and Ecovio/HP 90/10 D0.9, selected for comparison.

The spectrum of the pure Ecovio D0.9 film shows the main characteristic peaks of its constituents, PBAT and PLA: C–H band in the range 3000–2800 cm^−1^; C=O stretching vibrations located at around 1749 cm^−1^ (PBAT) and 1710 cm^−1^ (PLA); a complex multiple band at 1250–1050 cm^−1^ due to C–O in the ester linkage; a sharp peak at 720 cm^−1^ due to four or more adjacent methylene (–CH_2_–) groups; and bending vibrations of benzene substitutes in the range 900–700 cm^−1^ [28,29]. The incorporation of the hazelnut perisperm within the polymer matrix changes the profile of the C–H band, since the HP vibrations at 2920 cm^−1^ and 2850 cm^−1^ become progressively more pronounced at increasing filler content. However, none of the films’ spectra has new peaks or noticeable peak position shifts, thus indicating that no significant filler–polymer interactions arise in the blends. The same observations can be made from the spectra taken on all the other films.

The effect of the HP addition on the film morphology was investigated by means of DSC analysis. The results are reported in Figure 8 and Table 6.

The DSC thermogram of the neat Ecovio film shows several thermal transitions corresponding to its PBAT and PLA constituents [28,29,30]. In particular, the PBAT phase gives a glass transition at ca. −30 °C and two melting peaks around 50 °C and 121 °C, corresponding to the melting of the crystal phases of the butylene adipate (BA) units and butylene terephthalate (BT) units, respectively. The PLA phase has a glass transition at about 60 °C, which is shaped as an endothermic peak, since PLA presents enthalpic relaxation [21], and a melting peak at 150 °C. All these transitions are also present in the thermograms of both Ecovio/HP 95/5 and Ecovio/HP 90/10 blend films, with some minor modifications. In fact, the Tg and the melting enthalpies of PBAT slightly decrease at increasing HP content, probably due to a release of small amounts of low molecular weight constituents able to promote the PBAT chains mobility and to reduce its ability to crystallize. In contrast, the melting enthalpies of PLA increases significantly in blends, which suggests that the HP filler can act as nucleating agent for PLA, especially for the film at lower HP concentration (5%). Finally, the thermogram of the Ecovio/HP 90/10 film also shows an additional endothermic peak at ca. −10 °C, corresponding to the melting of the oily part of the hazelnut perisperm; this peak is not visible in the less loaded film sample.

Table 7 reports the results of the mechanical tensile tests and oxygen permeability measurements performed on the Ecovio film and on all the active films with different average thickness and dried hazelnut perisperm loading. The mechanical parameters of Ecovio are in the characteristic range of PLA and PBAT blends [28,31] and very close to the typical values of pure PBAT. This is an indication that this grade of Ecovio contains PBAT as the most abundant constituent. The addition of HP reduces the elastic modulus, the stress at yield and the stress at break values up to an extent of ca. 25% for the Ecovio/HP 90/10 film, whereas it significantly increases the elongation at break of ca. 65% for the Ecovio/HP 95/5 sample and ca. 45% for the Ecovio/HP 90/10 one. Several factors contribute to this behavior, including the chains mobility and the crystalline morphology of the polymer and the release of low molecular weight compounds from the filler to the matrix. Ecovio is a semi-crystalline polymer with a very complex crystalline morphology, and its high strength and ductility depend to a large extent on the crystalline structure developed inside the material during processing. As pointed out by DSC measurements, the addition of HP filler interferes with the regular arrangement of the material causing a small decrease of the PBAT crystallinity and a small increase of the PLA one. Moreover, the filler increases the polymer molecular mobility, as indicated by the slight lowering of the Tg values, due to a partial release of its oily low molecular weight constituents. These changes translate in lower stiffness and higher ductility of the films filled with HP. Then, as the film thickness decreases, there is a further gradual reduction of the film stiffness and also a gradual lowering of ductility, which still remains higher or at least comparable to that of pure Ecovio. This trend can be explained considering that as the film thickness decreases, the HP particles tend to agglomerate due to the enhancement of their interactions. Consequently, they not only become less effective in restricting the polymer molecular mobility, giving a tensile strength decrease, but they also act as defects causing stress concentration points and thus reducing elongation.

For what concern the permeability to oxygen, the pure Ecovio film has a P O_2_ value within the typical range of PLA/PBAT blends [28]. The addition of HP causes a worsening of the barrier performance of the film, which becomes more relevant as the HP loading increases and the film thickness decreases. This trend is similar to that of the elastic modulus, as expected. In fact, the barrier properties and the elastic modulus of a polymer system are correlated, being affected by the same chemical–morphological factors (e.g., interphase adhesion, crystallinity, molecular mobility, etc.).

To examine the radical scavenging activity (RSA) of the active films, preliminary DPPH tests were applied on Ecovio/HP 95/5 and Ecovio/HP 90/10 samples with different thicknesses, maintaining the HP concentration constant at 100 μg/mL. The DPPH test was also carried out on pure Ecovio film that, as expected, showed no antioxidant activity. The results of the DPPH test are compared in Table 8.

All the films loaded with HP have antioxidant activity due to polyphenol content. Even if this activity is at least halved with respect to those of the pure hazelnut perisperm, the RSA% values are significant and included in the range 23–38, which makes these films useful in counteracting lipid oxidation and interesting from a commercial point of view for preserving the quality of lipophilic foods such as nut-dried fruits. Moreover, for both the Ecovio/HP 95/5 and Ecovio/HP 90/10 films, the activity is the highest for the thickest film and decreases with the film thickness, more strongly for the more loaded film. In fact, the difference is limited to ca. 2.5 percent units for the films at HP = 5 wt% and is ca. 13% for those at HP = 10 wt%. The decrease of RSA with the film thickness may be related to the worsening of the HP particles distribution into the polymer matrix occurring at increasing draw-up speed, as revealed by optical microscopy analysis. The agglomeration of the HP particles gives a lower exposed specific surface area that can account for a minor activity of the filler.

The films were tested for their optical characteristics, which have an important role for packaging applications since they provide see through property or allow blocking light radiation that promotes deterioration of packaged food. Table 9 reports the transparency (at 560 nm) values and the color parameters of the films at different compositions and draw-up speeds.

The table shows that all films have quite low transparency since the Ecovio matrix is mainly constituted of PBAT, which is a white and opaque polymer [30]. In particular, taking as a reference the pure Ecovio D.09 film, the transparency is almost completely lost after the addition of HP in films Ecovio/HP 95/5 D0.9 and Ecovio/HP 90/10 D0.9, produced at the same draw-up speed, due to the opacity of the hazelnut perisperm. However, at fixed composition and at increasing draw-up speed (i.e., at decreasing film thickness), the transparency gradually improves up to a maximum of 12.7% for the Ecovio/HP 95/5 D3 film.

The color of the films was evaluated in the CIELAB space through the measurements of the L*, a* and b* coordinates: L* indicates lightness and its values range from 0 (black) to + 100 (white), a* is the red/green coordinate (+ = red, − = green) and b* is the yellow/blue coordinate (+ = yellow, − = blue). Comparing the values of the coordinates, reveals that the addition of HP makes the films darker and modifies their color from white (a* and b* close to zero) to brown (+ red and + yellow). The effect increases with both HP loading and film thickness. The color difference ∆E changes correspondingly.

Finally, the films at different HP loadings were tested for their sealing ability to check possible benefits or negative effects of the perisperm addition on their weld resistance [32,33]. In particular, the films were welded at different temperatures and then submitted to delamination tests to measure the sealing strength. The results are compared in Figure 9. The graph shows that in all cases, the seal strength increases with the welding temperature. The pure Ecovio D0.9 film is weldable in a rather narrow temperature range, from 95 °C to 100 °C, whose upper limit is given by the melting of PBAT. At 90 °C, on the other hand, the temperature is not enough to soften the polymer; therefore, the film does not weld. Moreover, all other films are essentially not sealable at 90 °C.

The addition of the HP makes the active films sealable in a wider temperature range, up to 110 °C. The only exceptions are the Ecovio/HP 95/5 D3 and Ecovio/HP 90/10 D3 films, which tend to break down at sealing T > 95 °C due to their low thickness. In the other cases, the seal strength increases with T in a measure depending on both HP loading and film thickness. Up to 100 °C, the seal strength remains always lower than that of the pure Ecovio D0.9, but at higher T, the Ecovio/HP 95/5 D1.6 and Ecovio/HP 90/10 D1.6 samples showed improved seal strength of ca. 40% and 60%, respectively. Moreover, other authors reported increments in the seal strength of biodegradable packaging materials by the addition of different types of functional fillers [32].

## 4. Conclusions

Eco-friendly active packaging films were successfully developed incorporating hazelnut perisperm into a biodegradable polymer matrix. Hazelnut perisperm, which is currently a waste product of the hazelnut production chain, was selected as active agent thanks to its high content of tocopherols. The films were produced from melt compounded polymer/HP blends at different composition, using a lab scale film-blowing extrusion plant.

A preliminary characterization of HP samples demonstrated that the hazelnut perisperm has high antioxidant activity, since at a concentration of 100 μg/mL it showed a DPPH free radical scavenging activity of about 85%, and is stable in the processing conditions required for film production. Films with different HP loadings and thicknesses were produced and tested for their morphology, functional properties of main interest for food packaging applications and radical scavenging activity.

In terms of morphology, ATR−FTIR and DSC analyses evidenced that the filler does not establish noticeable interactions with the polymer but causes a slight increase of the polymer molecular mobility and modifies its crystalline morphology. One cause of these phenomena is the partial release by the HP filler of its oily constituents into the polymer matrix. Consequently, the films added with HP show lower stiffness, higher ductility, and higher permeability to oxygen with respect to the pure Ecovio D0.9 one. The changes are more relevant at HP loading increases and film thickness decreases. In terms of optical appearance, all films have the characteristic opacity of the pure Ecovio matrix but, while this one is white, all active films acquire the brownish color of the HP filler, with a higher color difference the higher the HP content and thickness. Regarding the sealing ability, the addition of HP gives films an extended sealability temperature range and, at optimized composition and film thickness, improvement in the seal strength up to ca. 60%.

Finally, preliminary DPPH tests carried out on all produced films showed that the incorporation of the hazelnut perisperm into the Ecovio polymer matrix is able to give relevant radical scavenging activity to the films, dependent on film composition and thickness and on HP distribution quality. The best performing system was the Ecovio/HP 90/10 D0.9, which exhibited RSA = 38%. The reduction of the RSA values with the HP distribution quality into the polymer matrix highlights the importance of optimizing the film production process to maximize the radical scavenging activity of the developed films.

In conclusion, these results open new possibilities for the development of circular and fully sustainable active packaging through the conversion of a food waste such as the hazelnut perisperm into a higher value product that can contribute back to the food supply chain. The proposed solution, in particular, has great potential in preserving the quality of nut-dried fruits such as hazelnuts. 

## Figures and Tables

**Figure 1 polymers-14-04156-f001:**
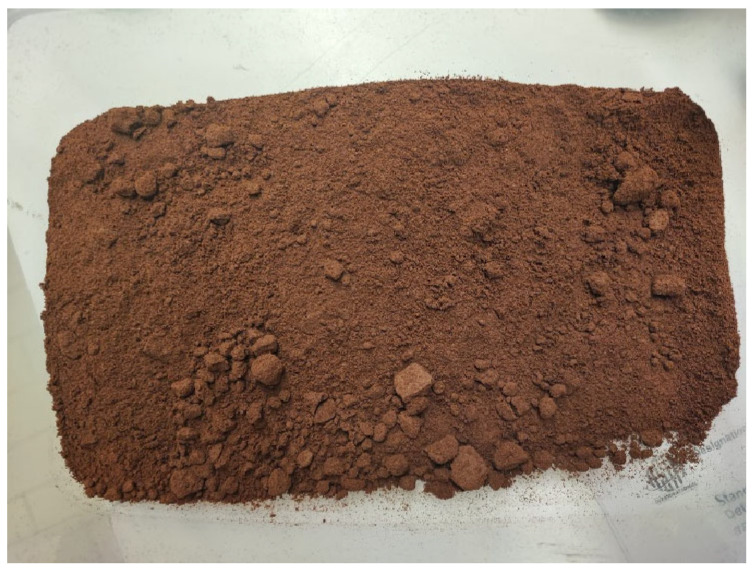
Appearance particle size of hazelnut perisperm powder (<0.25 mm).

**Figure 2 polymers-14-04156-f002:**
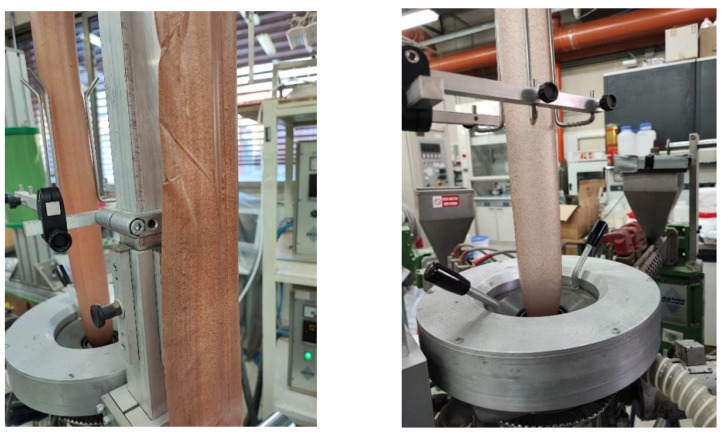
Film blowing of Ecovio/HP 90/10 at different draw-up speeds:1.6 m/min (**left**) and 3 m/min (**right**).

**Figure 3 polymers-14-04156-f003:**
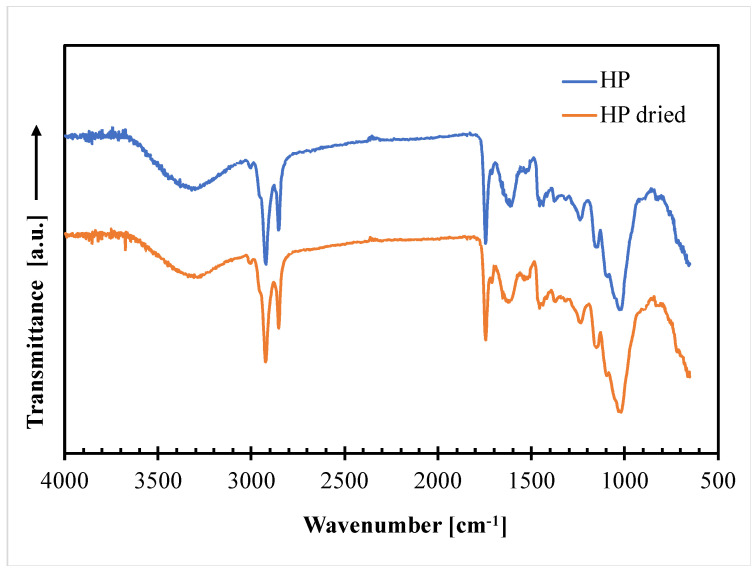
ATR−FTIR curves of not dried (HP) and dried (HP dried) hazelnut perisperm samples.

**Figure 4 polymers-14-04156-f004:**
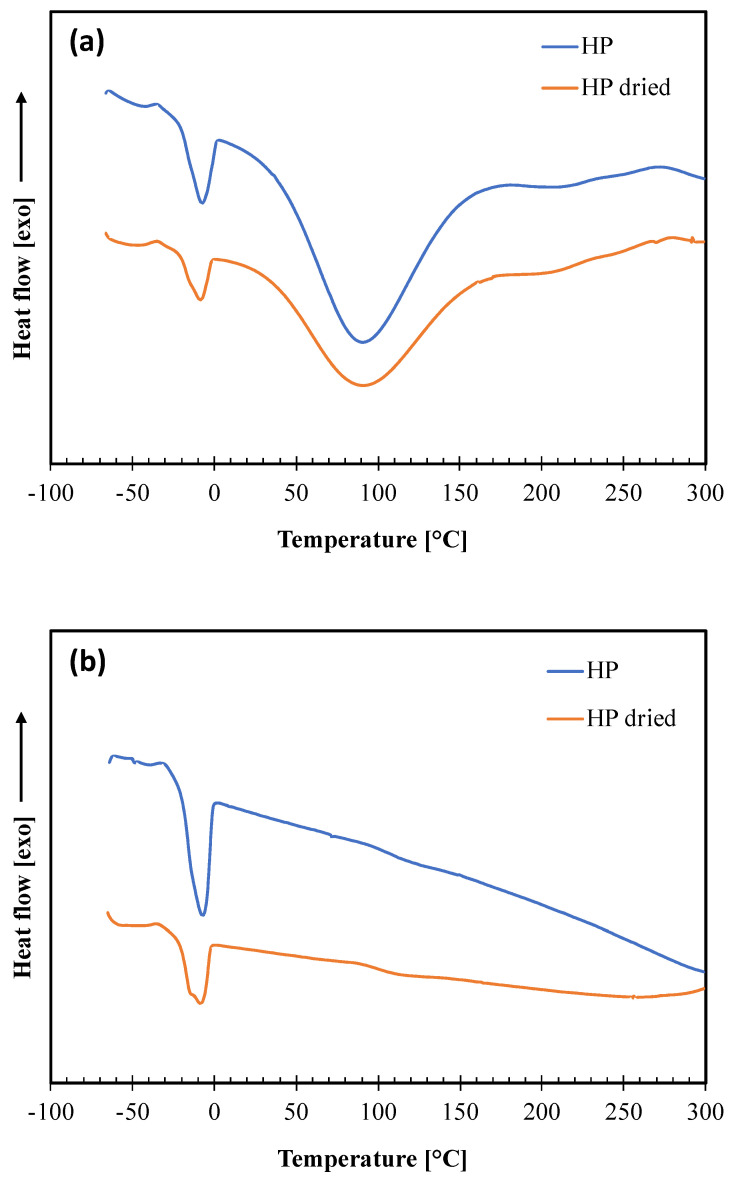
DSC thermograms of not dried (HP) and dried (HP dried) hazelnut skin samples: (**a**) first heating scan; (**b**) second heating scan.

**Figure 5 polymers-14-04156-f005:**
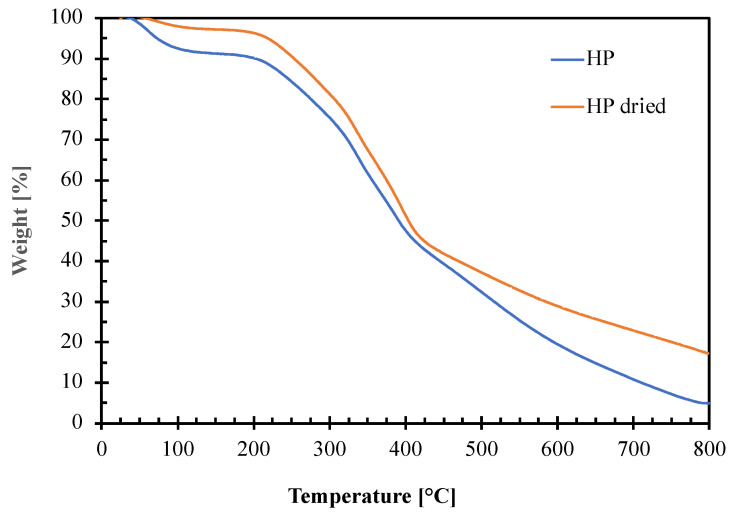
Thermogravimetric curves of not dried (HP) and dried (HP dried) hazelnut perisperm samples.

**Figure 6 polymers-14-04156-f006:**
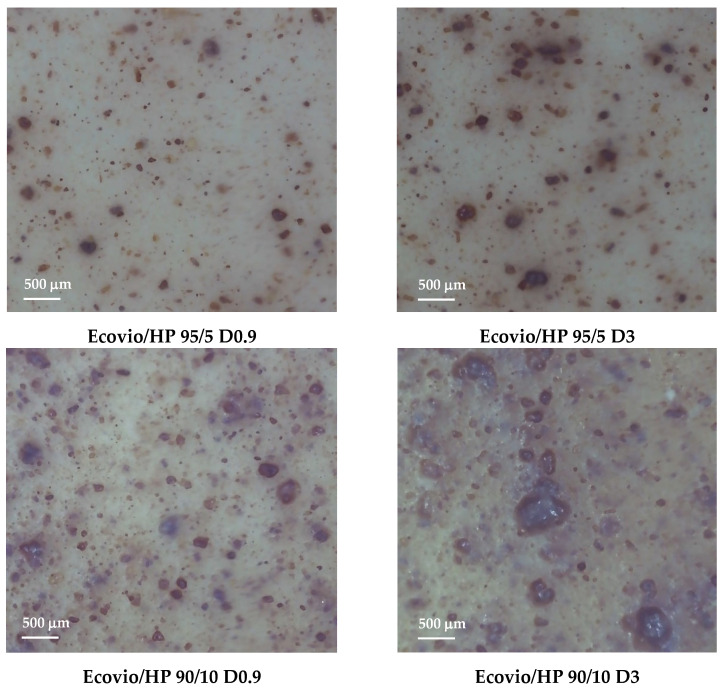
Optical microscopy images of Ecovio/HP 95/5 and Ecovio/HP 90/10 active films produced with different draw-up rates.

**Figure 7 polymers-14-04156-f007:**
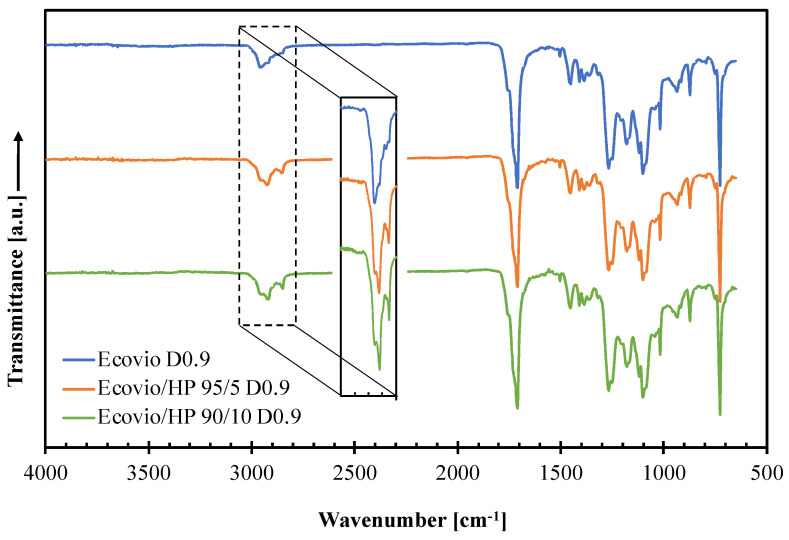
ATR−FTIR curves of Ecovio films produced at different HP loadings (0, 5, 10 wt%) and draw-up speed equal to 0.9 m/min (D0.9).

**Figure 8 polymers-14-04156-f008:**
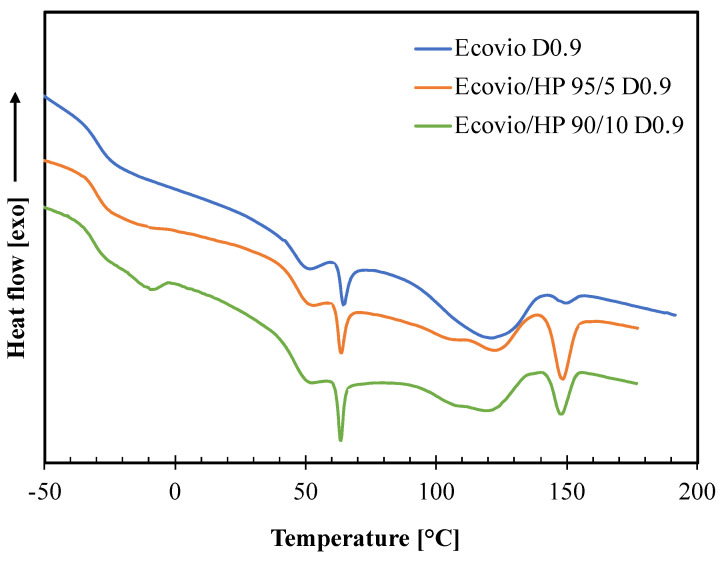
DSC thermograms of active films produced at different HP loadings (0, 5, 10 wt%) and draw−up speed equal to 0.9 m/min (D0.9).

**Figure 9 polymers-14-04156-f009:**
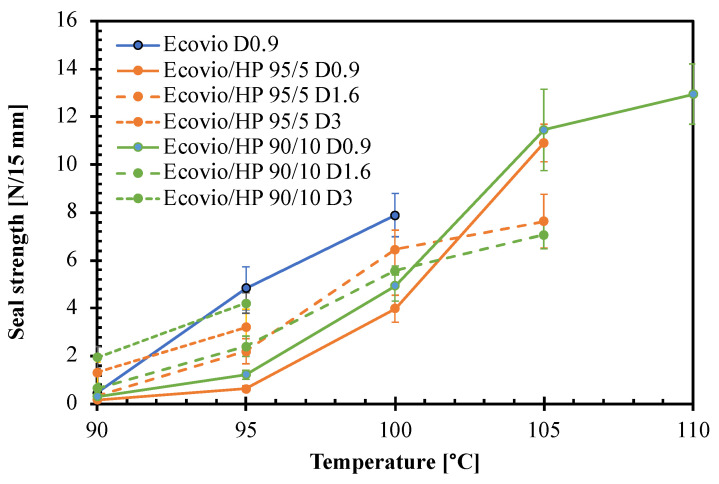
Effect of welding temperature on sealing strength of films with different average thickness and hazelnut perisperm loading.

**Table 1 polymers-14-04156-t001:** Typical basic material properties of Ecovio F2332.

Property	Test Method	Ecovio F2332
Mass Density [g/cm^3^]	ISO 1183	1.24–1.26
Bulk Density [kg/m^3^]	DIN EN ISO 60	750
Melt Volume Rate @ 190 °C, 5 kg [mL/10 min]	ISO 1183	7.0–11.0
Melting points [°C]	DSC	110–120140–155

**Table 2 polymers-14-04156-t002:** Nomenclature and average thickness of film samples produced at different draw-up speeds.

Sample Name	Draw-Up Speed [m/min]	Average Film Thickness [μm]
Ecovio D0.9	0.9	55 ± 4
Ecovio/HP 95/5 D0.9	0.9	120 ± 09
Ecovio/HP 95/5 D1.6	1.6	90 ± 7
Ecovio/HP 95/5 D3	3	65 ± 6
Ecovio/HP 90/10 D0.9	0.9	136 ± 6
Ecovio/HP 90/10 D1.6	1.6	104 ± 8
Ecovio/HP 90/10 D3	3	78 ± 5

**Table 3 polymers-14-04156-t003:** DSC thermal parameters of not dried (HP) and dried (HP dried) hazelnut skin samples.

Sample	1st Heating				Cooling		2nd Heating	
	Tm I [°C]	ΔHm I [J/g]	Tm II [°C]	∆Hm II [J/g]	Tc [°C]	∆Hc [J/g]	Tm I [°C]	ΔHm I [J/g]
HP	−7.6	16.8	90.4	262.2	−53.3	7.9	−7.7	11.9
HP dried	−8.4	9.8	91.2	209.8	−58.0	4.5	−8.5	7.7

**Table 4 polymers-14-04156-t004:** Degradation onset temperatures (T_onset_) and DTGmax temperatures at which the degradation speed is maximum for not dried (HP) and dried (HP dried) hazelnut perisperm samples.

Sample	T_I_ [°C]	Weight Loss at T_I_ [%]	T_onset_ [°C]	T_II_ [°C]	T_III_ [°C]	T_IV_ [°C]
HP	63.0	8.9	214.7	275.0	340.1	388.4
HP dried	71.6	3.0	208.5	275.9	340.0	398.0

**Table 5 polymers-14-04156-t005:** Radical scavenging activity (RSA%) of not dried (HP) and dried (HP dried) hazelnut perisperm samples.

Sample	RSA [%]
HP	85.7 ± 1.1
HP dried	83.2 ± 1.3

**Table 6 polymers-14-04156-t006:** DSC results of active films produced at different HP loadings (0, 5, 10 wt%) and draw-up speed equal to 0.9 m/min (D0.9).

Film Sample	Tg_PBAT_ [°C]	Tm_oil_ [°C]	ΔHm_oil_ [J/g]	Tm_BA_ [°C]	ΔHm_BA_ [J/g]	Tg_PLA_ [°C]	Tm_PBAT_ [°C]	ΔHm_PBAT_ [J/g]	Tm_PLA_ [°C]	ΔHm_PLA_ [J/g]
Ecovio D0.9	−29	-	-	50	1.1	64	121	6.4	150	0.3
Ecovio/HP 95/5 D0.9	−30	-	-	51	0.9	63	123	5.4	148	2.5
Ecovio/HP 90/10 D0.9	−31	−9.4	0.8	50	0.9	63	122	5.5	148	1.5

**Table 7 polymers-14-04156-t007:** Tensile mechanical parameters and permeability to oxygen of films with different composition and average thickness.

Film Sample	E [MPa]	σy [MPa]	σb [MPa]	εb [%]	P O_2_ [cm^3^·mm/(m^2^·d·bar)]
Ecovio D0.9	157 ± 2	8.66 ± 0.69	9.96 ± 1.18	308 ± 25	41.0 ± 0.3
Ecovio/HP 95/5 D0.9	125 ± 4	6.49 ± 0.75	7.47 ± 1.48	507 ± 42	53.2 ± 1.0
Ecovio/HP 95/5 D1.6	112 ± 8	4.86 ± 0.34	6.92 ± 1.39	390 ± 56	64.5 ± 1.6
Ecovio/HP 95/5 D3	76 ± 5	4.75 ± 0.41	5.15 ± 0.58	273 ± 31	96.4 ± 0.5
Ecovio/HP 90/10 D0.9	118 ± 6	6.64 ± 1.01	7.94 ± 1.26	442 ± 40	56.4 ± 0.3
Ecovio/HP 90/10 D1.6	96 ± 1	5.38 ± 0.57	6.27 ± 1.23	395 ± 45	86.1 ± 2.4
Ecovio/HP 90/10 D3	52 ± 6	3.91 ± 0.56	4.06 ± 0.55	270 ± 25	135.5 ± 1.0

**Table 8 polymers-14-04156-t008:** Radical scavenging activity (RSA%) of Ecovio/HP 95/5 and Ecovio/HP 90/10 active films with different average thicknesses.

Film Sample	RSA [%]
Ecovio D0.9	n.d.
Ecovio/HP 95/5 D0.9	25.4 ± 1.3
Ecovio/HP 95/5 D1.6	25.0 ± 1.1
Ecovio/HP 95/5 D3	23.0 ± 1.4
Ecovio/HP 90/10 D0.9	38.2 ± 1.3
Ecovio/HP 90/10 D1.6	28.6 ± 1.4
Ecovio/HP 90/10 D3	25.0 ± 1.3

**Table 9 polymers-14-04156-t009:** Transparency and CIELAB color variables of films with different average thickness and hazelnut perisperm loading.

Film Sample	Transparency at 560 nm [%]	CIELAB Coordinates			
		L*	a*	b*	∆E
Ecovio D0.9	5.3 ± 1.0	96.5 ± 0.1	−0.8 ± 0.1	2.5 ± 0.1	0
Ecovio/HP 95/5 D0.9	0.9 ± 0.1	66.8 ± 2.0	9.3 ± 0.6	16.9 ± 0.2	34.5
Ecovio/HP 95/5 D1.6	3.1 ± 0.5	76.5 ± 1.6	6.0 ± 0.5	15.0 ± 0.4	24.6
Ecovio/HP 95/5 D3	12.7 ± 1.8	85.9 ± 1.3	2.8 ± 0.4	10.5 ± 0.7	13.8
Ecovio/HP 90/10 D0.9	0.4 ± 0.2	49.8 ± 1.2	12.9 ± 0.1	16.4 ± 0.9	50.6
Ecovio/HP 90/10 D1.6	3.2 ± 0.7	64.8 ± 1.5	9.4 ± 0.4	18.3 ± 0.1	36.9
Ecovio/HP 90/10 D3	9.2 ± 2.6	77.1 ± 1.9	5.6 ± 0.6	14.6 ± 0.8	23.8

## Data Availability

The data presented in this study are available on request from the corresponding author.

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
