# Peer review of "Use of Hazelnut Perisperm as an Antioxidant for Production of Sustainable Biodegradable Active Films"

_polymers, 2022, doi:10.3390/polym14194156_

Round 1
Reviewer 1 Report
The research appears to be well-planned, executed and presented. The only serious concern that I have is the identity of the tissue being analyzed. Botanically, the perisperm is a tissue associated with the endosperm in certain plant families, though my suspicion is that the Betulaceae, the family which contains hazelnut is not one of these families. Rather, the researchers are using the outer portion of the edible hazelnut. When this outer layer of the hazelnut seed originates from maternally derived tissue arising from the inner and outer ovular integuments, I believe it is referred to as a seedcoat. I believe this is the case for walnuts and so possibly hazelnuts. However, if the outer coat of the seed is derived from the outer ovary tissue, I believe it’s referred to as a pellicle, as is the case for almonds. The authors should identify the developmental origin of this tissue in order to use the proper botanical term in their publication to avoid confusion. The different origins [ovular integuments for seedcoat, outer ovary for pellicle, or perisperm (endosperm-like nutritive tissue in early embryo development)] also have important implications for the genetic (and so biochemical) uniformity of the material the authors are promoting. Because the ovular integuments are somatic tissue, all the tissue being processed would be genetically identical to the Tonda di Giffoni source variety and so would be genetically uniform (though some biochemical variation from environmental differences would be expected). If the tissue is truly endosperm/perisperm in origin, each hazelnut seed used in extraction could be genetically (and so biochemically) variable because of meiosis/mitosis. A pellicle origin would be genetically more ambiguous but probably intermediate to these two extremes.
For production uniformity, an ovular integuments origin is thus desirable (and very likely the case for hazelnuts). A more accurate identification of your source material would thus strengthen your research arguments.
Similarly, the term seed ‘peel’ should be avoided because the hazelnut also has a hull or fruit, the outermost layer of which is commonly referred to as the peel.
Author Response
I thank you the reviewer for the comment. The research was performed using the hazelnut perisperm as active ingredient. To clarify this, we replaced everywhere in the manuscript the terms "peels" and "skin" with the term "perisperm".
Reviewer 2 Report
The study was basically simple and clear. the authors should check out the typos throughout the manuscript (eg. it should be "HP" instead of "PH" in Figure 7), and also it would be perfect if the authors can share the optical pictures of the films.
Author Response
The manuscript was checked for typing and grammar errors. The legend of fig. 7 was corrected, too.
Reviewer 3 Report
The manuscript entitled "Use of hazelnut perisperm as an antioxidant for production of 2 sustainable biodegradable active films" obtained a very interesting results and open a new application for the hazelnut perisperm and interesting that they apply it by using the extrusion.
Some suggestions to improve the interesting manuscript:
1- Line 34 the Latin name of hazelnut nut should be Italic (Corylus avellana L.).
2- Line 53, 60 check how to write the reference in the beginning of the sentences
3- Line 63 the S. aureus, should be written all Staphylococcus aureus.
4- Line 326 the Table 8 is not for results of the mechanical tensile tests and oxygen permeability measurements. The correct table is Table 7.
5- Line 399 to 423 I suggest to compare your obtained results with the some published articles. I suggest to you
- Nanomaterials 2020, 10, 52; doi:10.3390/nano10010052,
- https://doi.org/10.1016/j.foodhyd.2018.08.008
- Polymers 2020, 12, 1613; doi:10.3390/polym12071613
6- Please add the significant symbols to your all Tables or Figures.
7- Line 330 ".. This is an indication that this grade of Ecovio contains mainly PBAT…) it is in the contrast of what mentioned in Line 85-86.
Kind Regards,
Author Response
1- Line 34 the Latin name of hazelnut nut should be Italic (Corylus avellana L.). The Latin name of hazelnut nut was set Italic (Corylus avellana L.).
2- Line 53, 60 check how to write the reference in the beginning of the sentences. The reference positioning was changed according to the journal editing rules.
3- Line 63 the S. aureus, should be written all Staphylococcus aureus. The S. aureus was written all Staphylococcus aureus.
4- Line 326 the Table 8 is not for results of the mechanical tensile tests and oxygen permeability measurements. The correct table is Table 7. The Table number was corrected.
5- Line 399 to 423 I suggest to compare your obtained results with the some published articles. I suggest to you
- Nanomaterials 2020, 10, 52; doi:10.3390/nano10010052,
- https://doi.org/10.1016/j.foodhyd.2018.08.008
- Polymers 2020, 12, 1613; doi:10.3390/polym12071613
Additional references (ref. 32, 33) have been added.
6- Please add the significant symbols to your all Tables or Figures. The significant symbols have been added to all Tables and Figures.
7- Line 330 ".. This is an indication that this grade of Ecovio contains mainly PBAT…) it is in the contrast of what mentioned in Line 85-86. Thank you for the comment, we have clarified this point specifying that "This is an indication that this grade of Ecovio contains PBAT as the most abundant constituent."
Round 2
Reviewer 3 Report
No comments.
The authors did all the comments and suggestion.
Regards,